# Pathophysiological Role and Diagnostic Potential of R-Loops in Cancer and Beyond

**DOI:** 10.3390/genes13122181

**Published:** 2022-11-22

**Authors:** Essak S. Khan, Sven Danckwardt

**Affiliations:** 1Posttranscriptional Gene Regulation, Cancer Research and Experimental Hemostasis, University Medical Center Mainz, 55131 Mainz, Germany; 2Institute for Clinical Chemistry and Laboratory Medicine, University Medical Center of the Johannes Gutenberg University, 55131 Mainz, Germany; 3Center for Thrombosis and Hemostasis (CTH), University Medical Center of the Johannes Gutenberg University, 55131 Mainz, Germany; 4German Consortium for Translational Cancer Research (DKTK), DKFZ Frankfurt-Mainz, 60590 Frankfurt am Main, Germany; 5German Center for Cardiovascular Research (DZHK), Rhine-Main, 55131 Mainz, Germany

**Keywords:** R-loop, co-transcriptional processing, transcription termination, (alternative) polyadenylation, human disease, biomarker, prognostic power

## Abstract

R-loops are DNA–RNA hybrids that play multifunctional roles in gene regulation, including replication, transcription, transcription–replication collision, epigenetics, and preserving the integrity of the genome. The aberrant formation and accumulation of unscheduled R-loops can disrupt gene expression and damage DNA, thereby causing genome instability. Recent links between unscheduled R-loop accumulation and the abundance of proteins that modulate R-loop biogenesis have been associated with numerous human diseases, including various cancers. Although R-loops are not necessarily causative for all disease entities described to date, they can perpetuate and even exacerbate the initially disease-eliciting pathophysiology, making them structures of interest for molecular diagnostics. In this review, we discuss the (patho) physiological role of R-loops in health and disease, their surprising diagnostic potential, and state-of-the-art techniques for their detection.

## 1. Introduction

R-loops are three-stranded structures also known as non-B DNA structures, which form when RNA hybridizes with a complementary DNA strand, resulting in DNA–RNA hybrids and a displaced non-template, single-stranded DNA (ssDNA) (Figure 1). While short DNA–RNA hybrids form transiently during transcription and lagging-strand DNA synthesis, R-loops differ from these structures. They span 100–2000 base pairs [1] and are typically located at the 5′ end of the elongating RNA polymerase II. R-loops are abundant in the mammalian genome [2]. They are frequently found in GC-rich regions where they play a multifunctional role in modulating diverse aspects of gene regulation including replication, transcription, transcription–replication collision, and epigenetics, as well as maintaining genome integrity [3,4]. Apart from their regulatory function, the accumulation of unscheduled R-loops can have a detrimental effect on genome integrity. They serve as a primer for aberrant replication, act as replication blocks and effectors of transcription stress, and function as a source of DNA damage, thereby disrupting genome integrity [4,5,6,7,8]. Hence, cells have evolved mechanisms that tightly control the formation and resolution of R-loops to ensure timely controlled gene regulation and to avoid undesired mutagenic events [5,9,10]. The abundance of R-loops is known to be regulated directly or indirectly by many proteins, mostly preventing RNA from hybridizing with DNA, thus reducing excessive R-loop accumulation [11]. Among these are proteins required for efficient transcription initiation [12], transcription elongation [13] and termination [14], polyadenylation [15], RNA splicing [16], and RNA packaging and export [17,18]. While substantial progress has been made to uncover the underlying mechanisms and biological role of R-loop formation (reviewed elsewhere in great detail [4,5,6,7,8]), increasing numbers of reports indicate the detrimental consequences of unscheduled RNA formation. For example, aberrant R-loop formation and accumulation has been associated with several human diseases [3,4,5,7,19], such as cancer [20,21,22] and neurological [23,24], hematological [25], and cardiovascular disorders [21,26]. In addition, the initial reports [27,28] indicate a potential role of R-loops as a potent diagnostic biomarker. In this review, we summarize the physiological role and pathophysiological consequences of R-loops and shed light on their diagnostic potential—a rapidly growing subject. We also discuss the current methodologies used to detect R-loops and future perspectives in this area.

## 2. Roles of Regulatory R-Loops in Gene Regulation and Genome Stability

For a long time, R-loops have been considered accidental by-products of transcription and a mere source of genomic instability when not removed correctly [5,9]. However, it has recently become evident that R-loops have various regulatory functions in biological processes. Depending on their role, R-loops are broadly classified as physiological or “regulatory” and pathological or “unscheduled” R-Loops [10] (Figure 1).

Regulatory R-loops are intermediates that modulate gene activity and the organization as well as stability of the genome. R-loops that are found in the promoter regions of genes are also known as *Promoter R-loops*. They actively regulate genes through several mechanisms [30,31,32,33,34]. They ease transcription by preventing the binding of DNA methylation enzymes (DNMT) [30,31]. They facilitate the binding of transcription factors [32] and prevent transcriptional repressors from binding to promote the transcription of genes [33]. In some cases, R-loops formed at the promoter region of transcription factors may also cause their silencing [34].

R-Loops are also enriched at the 3′ ends of some genes and aid transcription termination by promoting various mechanisms [9]. They promote the termination of backtracked Pol II [10]. They also trigger antisense transcription to reinforce RNAPII pausing by recruiting the RNA interference machinery [8,35]. R-loops are known to regulate the activity of chromatin remodelers [35,36,37,38]. They can induce chromatin de-condensation [36]. On the contrary, they can also promote heterochromatin assembly [35,37] and chromatin compaction [38]. Other than transcription, R-loops also facilitate DNA replication initiation in mitochondrial DNA [5], immunoglobulin (Ig) class switch recombination [39], or CISPR-Cas9 activity to facilitate Cas-9 mediated cleavage [40]. These few examples illustrate that regulatory R-loops have a plethora of functions by modulating diverse aspects including genomic structure, organization, and gene expression.

Apart from gene regulation, R-loops are also involved in maintaining the integrity of the genome. They promote DNA double-stranded breaks’ (DSB) repair, the repair of short telomeres, and preserve DNA topology [7,9]. The R-loops at DSBs develop from either de novo transcription from a free 3′ end at the break site or from transcription stalling within a gene [9]. Mechanistically, the R-loops formed at the DSBs attract breast cancer susceptibility protein 1 (BRCA1 [41]) and eventually other repair factors (BRCA2 [41]) to repair the DNA. Later, R-loops are removed by senataxin (SETX) [42], DEAD-box helicase 1 (DDX1) [43], ribonuclease (RNase) H1 [44] or RNase H2 [45], or homologous recombination [46] (further details on R-loop-modifying proteins are provided in Table 1 and Section 4). The accumulation of *telomere repeat-containing RNA* (TERRA) R-loops [47,48,49] at short telomeres [49] promotes DNA repair through RAD51-mediated homology-dependent repair [41,42,50]. Furthermore, R-loops are known to act as topological ‘‘stress relief valves’’ by relieving the higher energy of supercoiled DNA with a surprising capacity for relaxing tension in the genome (e.g., one R-loop can relax ~18 negative supercoils [51]) and can also extend to a variety of lengths beyond conserved sequences rich in CG or purines [52]. Taken together, regulatory R-loops are thus important for central aspects of genomic organization, gene regulation, and the maintenance of genome integrity.

## 3. Role of Unscheduled R-Loops as a Source of DNA Damage and Genomic Instability

Over the last few years, there has been increasing evidence that R-loops act as a double-edged sword [6,7,8]. Although R-loops are important regulatory elements for gene expression and genome stability, the aberrant or unscheduled formation of R-loops has been implicated in various diseases [6,7]. Unscheduled R-loop formation and R-loop accumulation can become a source of DNA damage. They are often linked to genome instability by forming weak ssDNA and DSBs [53] as well as potentiating replication fork stalling [54] and transcription–replication collision due to the low level or activity of topoisomerase 1 leading to recombination and genome instability [55,56].

The generation of DNA damage and replication stress triggers the activation of complex surveillance mechanisms, collectively called the “DNA damage response” (DDR), which are crucial to maintaining genome integrity and thus avoiding perturbations of a wide array of biological processes. Recent evidence suggests that unscheduled R-loop formation results in the activation of phosphatidylinositol 3-kinase-related protein kinases, including mammalian ATM (Ataxia–telangiectasia-mutated) and ATR (ATM- and Rad3-related) [57], which are key players governing the DDR. Both kinases are activated by DDRs, but their specificities are different. The dysfunction of R-loop homeostasis results in the activation of ATM/Tel1 and ATR/Mec1 kinase in a distinct manner. ATM/Tel1 is a multifunctional kinase that helps maintain genomic stability through its control of numerous aspects of cellular survival including telomere homeostasis [58,59]. To maintain the integrity of the genome, stalled replication forks are controlled by a checkpoint whose central player is the human kinase ATR/Mec1. It helps stabilize the replisome directly or by activating the checkpoint response to control DNA repair, fork restart, and other mechanisms for cell cycle progression [60]. Here, DNA damage due to unscheduled R-loop formation causes transcription–replication conflicts (TRCs) that trigger the ATR activation of the S-phase checkpoint [60]. Although many functional aspects of the underlying mechanisms remain to be elucidated, unscheduled or pathological R-loops are an important source of DNA damage and genome instability and can, therefore, become pathogenic (further detail is provided below). Thus, cellular mechanisms must be in place that control the formation and resolution of R-loops.

## 4. Role of R-Loop-Binding Proteins and (Co-)Transcriptional Mechanisms in R-Loop Formation, Resolution, and Prevention of Aberrant R-Loop Accumulation

Numerous proteins and almost all processes involved in gene regulation control the dynamics of R-loop formation directly or indirectly, thereby preventing excessive R-loop accumulation (Table 1). The latter is likely best explained by the fact that the coordinated co-transcriptional processing and packaging of the nascent transcript into ‘inert’ ribonucleoprotein particles (RNPs) ensures that R-loops occur in a scheduled manner, thus preventing the formation of otherwise deleterious DNA–RNA hybrid structures [61]. Events that perturb the coordinated co-transcriptional processing can result in unscheduled R-loop accumulation, leading to replication-associated DNA damage. Several examples illustrate this principle. These include events and proteins involved in transcription initiation [62], elongation [17,63], RNA splicing [16] and polyadenylation [15], RNA packaging and export [17,18], R-loop processing [64,65,66], and DNA topology [67]. For example, during transcription initiation, R-loops are formed by capping enzyme-RNA polymerase II complex (CE-Pol II) to facilitate efficient transcription [68]. Here, the capping enzyme is recruited on Pol II, which modulates the displacement of the nascent RNA to form R-loops and thus enables the co-transcriptional capping of the pre-mRNA and elongation [62]. This is followed by the recruitment of the *Facilitates chromatin transcription* (FACT) complex (a chromatin-reorganizing complex that swaps nucleosomes around the RNA polymerase during transcription elongation and replication). It promotes replication by acting as a histone deposition chaperone contributing to nucleosome assembly to regulate untimely R-loop-mediated TRCs [17]. Hence, the depletion of the FACT complex causes excessive R-loop accumulation and impairment in replication and transcription, illustrating a critical function in the resolution of R-loop-mediated TRCs [17]. In addition, after transcription initiation and elongation, mRNA cleavage and polyadenylation (CPA) factors contribute to maintaining genome integrity by suppressing R-loop formation and modulating efficient mRNA cleavage [15,29]. CPA constitute an essential step in preventing replication-stress-associated DNA damage as transcription hinders replication fork progression and stability, and vice versa, compensatory pausing of co-transcriptional CPA emerges as a conserved mechanism of the DNA damage response (DDR) to allow for DNA repair and to avoid the propagation of genomic mutations [29]. Hence, perturbations of CPA-mediated co-transcriptional RNA processing are detrimental and can result in the formation of unscheduled R loops and genome instability [29]. For example, a loss of function of the 3′ end cleavage and polyadenylation factors (including PCF11, CLP1, FIP1L1, CFT2, and WDR33) results in DNA damage [15]. This cause’s excessive unscheduled R-loop accumulation tightly associated with replication-stress-induced DNA damage, the inhibition of transcription-rescued fork speed, origin activation, and alleviated replication catastrophe [69]. However, further factors localized at the RNA 3′ end and cooperating in transcription termination show a similar functional outcome, as demonstrated by the loss of function of Rtt103, the yeast homolog of RPRD1B [15], SETX [70], and XRN2 [71].

Another functional example is the *RNA-processing and export* (THO/TREX) complex, which is also involved in R-loop regulation. THO/TREX, a conserved nuclear complex, functions in messenger ribonucleoprotein (mRNP) biogenesis [72]. It inhibits aberrant R-loop formation and prevents transcription-associated recombination [13,73]. Mutations of hpr1 (part of the THO complex) result in the genome-wide impairment of replication progression in transcribed genes due to extensive R-loop accumulation at the replication fork [73].

**Table 1 genes-13-02181-t001:** Identity and function of R-loop-binding and R-loop-regulating proteins.

Protein	Function
**Transcription initiation and capping**	
Capping enzyme-Pol II complex [62]	Responsible for transcription initiation by modulating displacement of nascent RNA during transcription, thereby promoting R-loop formation
**Transcription elongation**	
Facilitates Chromatin Transcription (FACT) complex [17]	Helps in preventing R-loop accumulation-causing TRCs
**Transcription termination, cleavage, and polyadenylation**	
Cleavage and Polyadenylation (CPA) factors [15,29,69] (PCF11, CLP1, FIP1L1, CFT2, WDR33)	Suppresses R-loop formation and facilitates efficient mRNA cleavage, thereby preventing replication-stress-associated DNA damage
**RNA processing and export**	
Transcription and export complex (THO/TREX complex; Tho2/THOC2, Hpr1/THOC1, Mft1, Thp2, Sub2/UAP56) [13,73]	Inhibits aberrant R-loop formation and transcription-associated recombination
**Splicing**	
Serine- And Arginine-Rich Splicing Factor SRSF2 [16]	Prevents the formation of mutagenic R loop structures
RNA-Binding Protein With Serine Rich Domain 1 (RNPS1) [16]	Forms complex with ASF/SF2 to prevent transcriptional R-loops
**R-loop degradation**	
RNase H1/2 [10,74]	Prevents aberrant R-loop formation by timely removal of these hybrids
**R-Loop-processing factors (DNA–RNA helicases)**	
Senataxin (SETX) [75]	Binds to replication forks to protect its integrity across RNA-Polymerase-II-transcribed gene and unwinds unnecessary R-loops
Aquarius (AQR) [76]	Prevents R-loop formation by unwinding DNA–RNA hybrids
DExH-Box Helicase 9 (DHX9) [77]	Prevents R-loop formation by melting DNA–RNA hybrid with a 3′–5′ polarity
DExH-Box Helicase 11 (DHX11) [78]	Converts RNA G-Quadruplex structures into R-Loops to promote IgH class switch recombination
Werner Syndrome RecQ-Like Helicase (WRN) [64]	Protects the replication fork by preventing unscheduled R-loop formation
**DNA topology**	
Topoisomerase I/IIIB [55,67,79]	Involved in maintaining R-loop resolution by interacting with RNA-splicing and DNA-processing factors
**DNA repair and genome maintenance**	
Ataxia Telangiectasia Mutated (ATM)/Ataxia Telangiectasia And Rad3 Related (ATR) Kinase [57]	DNA-damage response (DDR) kinases that become activated when R-loop-mediated DNA damage occurs
Breast Cancer Type 2 Susceptibility Protein (BRCA2/FANCD1) [41]	Binds to R-loops in response to dsDNA breaks to invite other DNA repair factors

The improper dissolution of co-transcriptionally formed R-loops constitutes potential roadblocks for transcription and enhances transcription-associated recombination events [10]. The intricate coupling between RNA processing, R-loop formation, and genome integrity also manifests in defects of RNA splicing and RNA degradation. During splicing, RNA *Serine/arginine-rich splicing factors* (SRs) such as the ASF/SF2 protein are recruited to nascent transcripts by RNA polymerase II, thus preventing the formation of mutagenic R loop structures [16,80]. The depletion of ASF/SF2 was shown to result in a single-stranded non-template strand of a transcribed gene due to the formation of aberrant R-loop structures, giving rise to genomic instability [16]. ASF/SF2 depletion-induced genomic instability can be alleviated by the overexpression of the *RNA-binding protein with Serine Rich Domain 1* (RNPS1) that has been suggested to function together with ASF/SF2 to form RNP complexes on nascent transcripts, thereby preventing the formation of transcriptional R-loops [16]. Along with splicing factors, RNA-degrading enzymes such as RNase 1 and 2 are also involved in regulating R-loops [10,74]. They eliminate hybrids created accidentally during replication, thereby suppressing genome instability associated with R-loop formation [15,81,82]. Consequently, the mutation of RNA-degrading enzymes has been shown to increase the formation of hybrids and associated genome instability [18].

Apart from the RNA-processing machinery, DNA–RNA hybrid-processing proteins such as DNA/RNA helicases are also involved in regulating R-loop resolution and processing. Popularly known as R-loop-processing factors, SETX [75], Aquarius (AQR) [76], DExH-Box Helicase 9 (DHX9) [77], DExH-Box Helicase 11 (DHX11) [78], and Werner Syndrome RecQ Like Helicase (WRN) [64] are DNA/RNA helicases, which are also involved in relieving replication stress. SETX associates with replication forks to protect its integrity across RNA-Polymerase-II-transcribed genes and unwinds unnecessary R-loops [75]. AQR prevents R-loop formation by constantly unwinding the DNA–RNA hybrid [76]. DHX9, belonging to the SF2 superfamily of nucleic acid-unwinding enzymes, melts DNA–RNA strands with a 3’–5’ polarity, thus contributing to transcriptional activation and thereby maintaining genomic stability [77]. DHX11, an RNA helicase that converts RNA G-Quadruplex structures into R-loops, promotes IgH class switch recombination [78]. WRN, which belongs to the RecQ family of helicases (RecQ helicases are an important family of genome surveillance proteins conserved from bacteria to humans often referred to as ‘guardians of the genome’), is involved in multiple pathways of DNA repair and the maintenance of genome integrity [83]. It protects the replication fork by forming a complex with *Werner helicase-interacting protein 1* (WRNIP1) and preventing unscheduled R-loop formation [64]. While DHX9 behaves similarly to WRN to unwind with a 3’–5’ polarity, DHX9 is considerably faster than WRN in unwinding RNA hybrids [64]. WRN preferably unwinds RNA-containing Okazaki fragment-like substrates whereas DHX9 fails to bind in order to unwind Okazaki fragment-like hybrids, suggesting a role in the lagging strand maturation of DNA replication. The depletion or mutations of all these DNA/RNA helicases cause aberrant, unscheduled R-loop accumulation leading to dsDNA breaks [64,65,66]. The exposed ssDNA acts as a source of DNA damage [65,66,77], ultimately causing impairment in replication and the transcription-associated recombination of cells.

Finally, R-loops can also be affected by the DNA topology itself, wherein topoisomerase I and IIB are involved in modulating R-loop dynamics. Topoisomerase I (Top1) acts at the interface between DNA replication, transcription, and mRNA maturation. It prevents replication fork collapse by suppressing the formation of R-loops by interacting with ASF/SF2 [55]. The loss of Top1 promotes R-loop formation, especially in the 18S 5′ region of the ribosomal DNA, imposing persistent transcription blocks when RNase H is limited [56]. Topoisomerase IIIB (Top3B) is a component of the *Tudor domain-containing protein 3* (TDRD3) complex, which relaxes negatively supercoiled DNA and reduces transcription-generated R-loops [67,79]. The loss of function of the TDRD3 complex leads to an increased accumulation of R-loops resulting in abnormal chromosomal translocations of genes [84].

Altogether, a plethora of R-loop-binding or R-loop-modulating proteins reflect dedicated functions in RNA and DNA metabolism as well as R-loop organization and processing. However, there are also R-loop-binding proteins that have functions beyond these biological processes [6,28]. As one would conclude from the information above, interference with DNA, RNA, and R-loop processing, for example, by altering the abundance of various facets of transcriptional, co-, and post-transcriptional gene regulation; epigenetic modifications; and further processes (such as RNA splicing, trafficking, and transcription, see Table 2), controls and sometimes disrupts the abundance of transcript isoforms encoding R-loop-binding proteins [29,85,86]. For example, key components pervasively regulating CPA and alternative polyadenylation (APA) [87] (including RNA-processing factors involved in the coupling of transcription termination and CPA such as PCF11 [85]) control the processing of various components involved in the formation and resolution of R-loops [87]. This includes established components directly involved in the resolution of R-loops such as RNAse H1, RNAse H2, the DNA–RNA helicase, DDX5/Ddp2, or AQR, but also exosome components with a similar role (EXOSC3/hRrp40, EXOCS4/hRrp41, or EXOCS6/hMtr3) [88]. Conversely, the loss of function of components involved in R-loop resolution (such as the R-loop-associated helicase, SETX) also affects APA [86]. This suggests that CPA and R-loops bi-directionally affect each other [29]. This corresponds to the two-sided nature of R-loops, wherein physiological (‘scheduled’) R-loops tune gene expression (including transcription termination and 3′ end processing), while pathological (‘unscheduled’) R-loops impair genome integrity, which is typically followed by an inhibition of CPA to limit the ‘release’ of emerging faulty transcripts and to allow for the repair of the genomic lesion [29]. Overall, this reflects the intricate network of co- and posttranscriptional gene regulation processes [61] involving R-loop dynamics, and provides an explanation for why aberrant/perturbed R-loop formation is frequently found in numerous disorders [4,5,6,7,8].

## 5. R-Loops Associated with Human Disease

The integrity of the genome and gene regulation is intensely controlled by scheduled R-loops (Figure 1). In contrast, unscheduled R-loops represent a source of DNA damage and genome instability. Hence, it is not surprising that unscheduled R-loops are increasingly linked to disorders. This includes disease entities where genome instability is an inherent element of the underlying pathophysiology (such as cancer) but also extends to other disorders including neuro-pathologies, where the intrinsic cell/tissue repair capacity is limited and hence perturbations of basic biological mechanisms are more likely to become visible. In the following, we present a few examples that document the stunning spectrum of such disorders associated with R-loops (Table 3).

**A**. 
**R-loops in Nucleotide Expansion Diseases**


About one million short tandem repeats are present in the human genome. These repeats are essential for genome integrity and function [90]. Unwanted expansions in DNA short tandem repeats give rise to so-called *Nucleotide Expansion Diseases* [91]. DNA repeats vary in size from dodecamers to longer, and the threshold at which these repeats expand to become symptomatic depends on the disease [92]. Over fifty human disorders are known [90]. A significant number of these expansions cause aberrant R-loop formation. This has been linked to common genetic disorders such as amyotrophic lateral sclerosis (ALS), where expanded hexanucleotide GGGGCC repeats are found in C9orf72 and ATXN [93,94], frontotemporal dementia (FTD, with hexanucleotide GGGGCC repeats in C9orf72) [94], polyglutamine-associated ataxias [95], spinocerebellar ataxias (SCAs, with an (CAG)n nucleotide expansion in ATXN1/2), Huntington’s disease, and Friedreich’s ataxia (GAA or TTC) [96,97,98,99]. R-loop accumulation is also seen in other nucleotide expansion diseases such as myotonic dystrophy (displaying CAG/CTG expansions) [100] and intellectual disability disorders such as Fragile X syndrome [101] (CGG-repeat-containing alleles of the FMR1 gene [102]). These examples suggest that alterations in cis that result in enhanced R-loop formation can be associated with various disorders, preferentially affecting neuronal cells.

**Table 3 genes-13-02181-t003:** R-Loop-linked diseases and genes associated with aberrant R-loop accumulation.

Diseases	Genes Associated with R-Loops
Aging [8,103,104]	SETX
Alzheimer’s [8,103,104,105]	SETX, WW domain-containing oxidoreductase
Aicardi–Goutières syndrome (AGS) [106]	TREX1, RNASEH2
AIDS-associated malignancies [107]	TREX complex
Amyotrophic lateral sclerosis (ALS) [30,93,94]	C9orf72 and ATXN2 (GGGCCC)n, SETX
Alternative lengthening of telomere (ALT)-dependent cancers [108]	TERRA complex
Ataxia with oculomotor apraxia (AOA2) [7,109,110]	SETX
Breast cancer [111,112,113,114]	BRAC1, BRAC2, Estrogen, SETX
Burkitt’s lymphoma [84]	c-MYC, TRD3-TOP3B
Colon cancer [115,116]	VIM
Myotonic dystrophy type 1 (DM1) [100]	DMPK
Embryonal tumors with multilayered rosettes (EMTR) [27]	C19MC
Eosinophilic leukemia [15]	FIP1
Ewing’s sarcoma [117]	EWS-FLI, BRCA1
Frontotemporal dementia (FTD) [94]	C9orf7 (GGGCCC)n
Fragile X syndrome type E (FRAXE) [101,102]	FRM2 (CCG)n
Friedreich ataxia (FRDA) or fragile X syndrome type A (FRAXA) [96,97,98,99]	FXN (GAA)n, FRM1 (CCG)n
Huntington’s disease (HD) [96,97,98,99]	HTT (CAG)n
Infertility [118]	SETX
Multiple myeloma [84,119]	c-MYC, TRD3-TOP3B, IFN
Myelodysplastic syndromes [120]	U2AF1 (S34F), SRSF2
Polyglutamine-associated ataxias [95]	Multifactorial Nucleotide Expansion disorder
Parkinson’s disease [8]	SETX
Spinocerebellar ataxias (SCAs) [39]	ATXN1/2 (CAG)n
Immunodeficiency, centromere instability, and facial anomalies (ICF) syndrome [121]	TERRA
Wiskott–Aldrich syndrome (WAS), X-linked thrombocytopenia (XLT), and X-linked neutropenia [25]	XLT-WAS

**B**. 
**R-loops in Neuronal Diseases**


In addition, to repeat nucleotide expansions, alterations in trans (in R-loop modifying components, such as SETX [35] or the THO/TREX complex [7]) can lead to aberrant R-loop accumulation and result in similar phenotypes. For example, recessive mutations in senataxin (SETX, a protein that protects the replication fork across the RNA-Polymerase-II-transcribed gene and unwinds unnecessary R-loops) cause R loop accumulation [35] in a particular type of ALS (ALS4) and Ataxia with oculomotor apraxia type 2 (AOA2) [7,109,110]. These mutations are also linked to aging and other neurodegenerative disorders, such as Parkinson’s and Alzheimer’s diseases [8,103,104].

In Aicardi–Goutières syndrome (AGS), mutations in RNASEH2 and TREX1 result in an increase in R-loop accumulation [106]. This demonstrates that RNASEH2 and TREX1, acting as an exonuclease as part of the THO/TREX complex, can have a clinical impact on R-loop resolution [7]. By analogy, the downregulation of the WW domain-containing oxidoreductase (proteins that are responsible for regulating transcription–replication collisions and preventing unwanted R-loop accumulation) is responsible for Alzheimer’s disease, showing aberrant R-loop accumulation [105]. These examples highlight the functional importance of the components that are involved in the formation and resolution of R-loops.

**C**. 
**R-loops in Cancer**


Genome instability is a characteristic of most cancers. Hence, it is not surprising that a fraction of cancers including AIDS-associated malignancies [107] and *Alternative lengthening of telomere* (ALT)-dependent cancers [108] are linked to R-loop accumulation in *Telomeric DNA and long noncoding RNA* (TERRA). TERRA is a key mediator of the *Alternative Lengthening of Telomeres* (ALT) pathway due to a dysfunction of RNase H1 [107,108]. In Estrogen (ET)-enriched breast cancer (BC), R-loop accumulation is highly enriched at E2-responsive genomic loci, resulting in E2-dependent R-loop-driven DNA damage [111]. In other BC types, BRCA1/SETX complexes cause R-loop-driven DNA damage [112,113]. The depletion of BRCA2 increases R-loop accumulation [122], which acts as a chief source of replication stress and cancer-associated instability in BC [114]. In multiple myeloma and Burkitt’s lymphoma, the depletion of TDRD3 (a protein responsible for gene transcription by interacting with TopIIIB, see above) exhibits elevated R-loop accumulation at the c-MYC locus in B cells, resulting in DNA damage and frequent chromosomal translocations. In eosinophilic leukemia, the truncated CPA component of FIP1L1 (mRNA 3′ end-processing factor interacting with PAPOLA and CPSF1) results in R-loop accumulation, thereby causing DNA damage and chromosome breakage [15]. In Ewing Sarcoma (an aggressive pediatric cancer of the bone and soft tissue), alterations of damage-induced transcription by the EWSR1 protein cause increased replication stress due to elevated R-loop accumulation [117]. All these examples highlight that R-loop accumulation acts as a source of DNA damage and causes the dysfunction of gene regulation and genomic instability in various cancers.

**D**. 
**R-loops in other diseases**


Apart from neuronal disorders and various cancers, unscheduled R-loop accumulation is also found in other diseases. In Immunodeficiency, Centromere instability, and Facial anomalies (ICF) syndrome, cells that exhibit short telomeres and elevated TERRA levels are enriched with R-loops at telomeric regions throughout the cell cycle. These telomeric R-loop hybrids are associated with high levels of DNA damage at the chromosome ends [121]. In *Ataxia with oculomotor apraxia* (AOA2) patients (caused due to mutations in SETX, a component with RNA helicase activity responsible for resolving R-loops), R-loop accumulation-induced DNA damage in cells undergoing spermatogenesis has been shown to cause sterility in males [118]. In *Myelodysplastic Syndromes* (MDS), caused by splicing factor mutations in components such as SRSF2 and U2AF1, aberrant R-loop accumulation is linked to the compromised proliferation of bone-marrow-derived blood progenitors, a characteristic feature of MDS [120].

The above-mentioned examples and further disorders [4,5,6,7,8] associated with unscheduled R-loops highlight the importance of mechanisms that prevent the formation and/or promote the resolution of unscheduled R-loop structures. While R-loops are not necessarily the cause for all the disease entities described so far, they may perpetuate and even aggravate the initially disease-eliciting pathophysiology. In either case, the abundance and association of R-loops with disease make them interesting structures for diagnostics.

## 6. R-Loops as a Diagnostic Biomarker?

Emerging reports suggest that accumulating unscheduled R-loops can be used for diagnostics and stratifying patients. For example, unscheduled R-loop accumulation is considered a novel molecular defect that is causative of TH1 immunodeficiency and genomic instability in patients with *Wiskott–Aldrich syndrome* (WAS) [113] and WAS-related disorders such as *X-linked thrombocytopenia* (XLT) and *X-linked neutropenia* [123]. In addition, the accumulation of undesired R-loops has been suggested to represent a potential biomarker for determining the prognostic outcomes in the XLT-WAS clinical spectrum [25]. Whole-genome sequencing of 193 primary *Embryonal tumors with multilayered rosettes* (ETMRs) revealed that unscheduled R-loop structures are widespread across these tumors due to a loss of DICER1 function (DICER1 promotes transcription termination at sites of transcription–replication collisions with DNA damage [27]), causing genomic instability. The targeting of R-loops with topoisomerase and poly ADP ribose polymerase inhibitors showed promising treatment strategies for this deadly disease [27]. Aberrant, unscheduled R-loop accumulation in *Uterine fibroid* (UF) patients (benign monoclonal neoplasms of the myometrium, which represent the most frequent non-cutaneous tumors) was demonstrated to specify potentially malignant tumor progression in the dominant UF subtype [124] (Figure 2A). In a study on *Epidermal growth factor receptor variant III* (EGFRvIII)-positive glioblastoma (GBM) patients, increased R-loop accumulation and genome instability caused by replication stress have been suggested to be associated with tumor heterogeneity and allow for the stratification of patients for individualized therapeutic approaches [125] (Figure 2B). In *Multiple Myeloma* (MM), elevated levels of R-loops and the failure to resolve R-loops can cause the sustained activation of a systemic inflammatory response characterized by the interferon (IFN) gene expression signature, which results in a poor prognosis [126]. Along with an elevated expression of para-speckle genes, R-loops have been identified to correlate with MM’s progression [119]. Altogether, unscheduled R-loop accumulation could be used to identify distinct biological properties (heterogeneity) in tumors [119,124,125], can guide the monitoring of the severity of disorders [25], and has a strong prognostic potential.

Beyond the identification of unscheduled R-loops, the abundance of R-loop-binding proteins (Table 1) can also be exploited as a potential biomarker. In a study by Boros-Oláh. et. al. in 2019, a systematic pharmacogenomic analysis was performed to test the drugging and diagnostic potential of an R-loop and its binding proteins in 33 cancer types [28]. R-loop-binding proteins of various categories with defined R-loop functions were selected for detailed analyses in this study (including proteins such as AQR, ATXN1/2, BLM, BRCA1/2, BUB3, BUGZ (ZNF207), CARM1, DDX19A, DHX9, EWSR, FANCD2, FANCM, GADD45A, PIF1, PRMT1, RNASEH1, RNASEH2, RTEL1, SETX, SRSF1, SRSF2, THO/TREX, TOP1, TOP3B, and U2AF1). Among these, the mRNA expression of BUB3, DHX9, PRMT1, THOC4, THOC7, U2AF1, and ZNF207 (BUGZ) were found to increase in several primary tumors compared to healthy tissues, while SRSF1 (ASF/SF2) was downregulated in most cancers (except for acute myeloid leukemia—LAML). Moreover, the expression of some of these R-loop-binding proteins was found to be directly linked to patient survival in various cancers, e.g., a low level of RNASEH2A, THOC6, PRMT1, and PIF1 were associated with prolonged survival in mesothelioma patients (MESO), while a low FANCM mRNA level appeared advantageous for breast cancer survival (Figure 2C). In another study, the depletion of TDP-43 (an R-loop-binding protein involved in RNA processing and with structural resemblance to heterogeneous ribonucleoproteins) in *Amyotrophic Lateral Sclerosis* (ALS) was linked to increased numbers of R-loops and DNA damage, which has been reported as a potential guide for developing ALS therapies [127]. Altogether, these examples highlight that R-loops and their binding proteins could represent potentially helpful biomarkers.

## 7. Emerging Technologies for Detecting R-Loops

As R-loops and their binding proteins show promising diagnostic potential, it is important to discuss the state-of-the-art technologies that can be used to detect R-loops. In 1980, the first technique for identifying R-loops was established using an antibody (S9.6) recognizing DNA–RNA hybrids, which revolutionized the R-loop field [128]. DNA–RNA Immunoprecipitation (DRIP) using the S9.6 antibody was further developed in vivo to uncover the contribution of R-loops to biological processes [56,70,129,130,131]. With the advent of next-generation sequencing, several techniques now exist to map the distribution, size, and dynamic changes of R-loops.

Globally, wet-lab techniques detect the accumulation of R-loops by foot-printing or pull-down assays probed with the R-loop-binding S9.6 antibody (Table 4). DRIP sequencing (DRIP-seq) is a widely used technique for the genome-wide profiling of R-loops. It utilizes the sequence-independent but high structure-specificity and affinity of the S9.6 monoclonal antibody to capture R-loops for the large-scale parallel DNA sequencing of the genomic fragments containing DNA–RNA hybrids [31]. However, this technique has some limitations, including bias and resolution limits because of the fragmentation of the genome using restriction enzymes, limited strand sensitivity, and a decreased feasibility with respect to conducting a quantitative analysis to study the genomic distribution. In addition, it does not discriminate between the R-loop sequence and the surrounding elements. To address these limitations, variants of DRIP have been developed. S1-DRIP-seq uses sonication for fragmentation where the displaced ssDNA is removed prior to sonication using S1 nuclease [132]. Although S1-DRIP generates a readout in high resolution (compared to DRIP), the technique’s reproducibility can be sometimes challenging, which currently limits the ‘clinical’ use of this technique (the S1 nuclease is delicate, and the reaction is difficult to control to obtain reproducible data [133]).

To address both the strand specificity and resolution of DRIP-RNA-seq [146], DRIPc-seq [133] and RDIP-seq [134] are techniques that have been established wherein the RNA components of R-loops are sequenced instead of the DNA. BisDRIP sequencing (bisDRIP-seq) allows for researchers to distinguish between the loops themselves and the surrounding DNA. It uses bisulfite to selectively convert cytosine residues into uracil residues within genomic DNA regions that contain single-stranded DNA. BisDRIPseq thereby allows for the mapping of R-loops at a near-nucleotide resolution to identify single-stranded regions based on the preferential labelling of one strand of the DNA and the requirement that the labelling is transcription-dependent [135]. SMRF-seq (Single-Molecule R-loop Foot-printing) is a technique consisting of the modified bisulfite-based mapping of the extruded single-strand DNA in an R-loop adapted for single-molecule long-read sequencing [137]. qDRIP is a quantitative differential DNA–RNA immunoprecipitation method combining synthetic DNA–RNA hybrid internal standards with high-resolution, strand-specific sequencing. It avoids biases inherent to read-count normalization by accurately profiling signals in regions unaffected by transcription inhibition. It thereby provides accurate differential peak calling between perturbed versus control conditions to obtain previously unattainable biological insights [136].

Alternative approaches to S9.6-based methods are DRIVE-seq, R-ChIP-seq, and MapR, which take advantage of catalytically inactive RNase H binding to R-loops. DRIVE-seq is conceptually similar to DRIP-seq as it utilizes the pull-down of hybrids through catalytically inactive RNase H instead of S9.6 [31]. R-ChIP is another method, and it can be used for the genome-wide profiling of R-loops and is designed to sequence the 5ʹ end of the template strand DNA, thus making the analysis different from that of typical ChIP-seq [138]. Both bis-DRIP and R-CHIP require an in situ step and show highly concentrated signals at the promoters of genes, but barely any signal at the 3′ end of genes. Other variants of DRIP show a strong signal at the promoter’s position and an appreciable signal in the gene body and at termination regions [33,110]. It is under debate whether the in situ step-containing technique is less sensitive to other regions or whether other variants of DRIP yield artifactual or non-specific signals [9]. Finally, Map-R is a fast, antibody-independent R-loop-profiling technique that utilizes RNase H to guide micrococcal nuclease to R-loops, which are subsequently cleaved, released, and identified by sequencing. It provides an output allowing for genome wide coverage with a low level of input material in a fraction of the time and with high sensitivity [139].

In addition, R-loop CUT&Tag sequencing has been established as a sensor-based technique used to overcome large discrepancies in R-loop mapping so as to provide an accurate and comprehensive profile of native R-loops across the genome. Some of its discrepancies arise due to fragmentation bias by restriction enzymes, disparate specificities of RNase H1 and S9.6 to R loops, or differences in sequencing and capture strategies, such as R-loop capture ex vivo or in situ. Moreover, full-length recombinant catalytically inactive RNase H1 is not very efficient in affinity pulldowns [8,12], which is the principle of DRIP-related R loop-mapping methods (see above). Therefore, R-loop-mapping methods that are independent of S9.6 or catalytically inactive RNase H1 are urgently needed to resolve controversies. R-loop CUT&Tag combines CUT&Tag and GST-His6-2×HBD (glutathione S-transferase–hexahistidine–2× hybrid-binding domain) tags as an artificial R-loop hybrid sensor to specifically recognize DNA–RNA hybrids. It is sensitive and generates good resolution for sensing the R-loop as compared to capture strategies that largely contribute to disparities in the previous techniques [140].

Very few techniques are available for characterizing both R-loops and their binding proteins. Recently, R-loop proximity proteomics has been developed to identify proteins that bind to R-loops and regulate them. RNA–DNA Proximity Proteomics (RDProx) is a technique that enables the mapping of an R-loop using the fusion protein of the hybrid-binding domain (HBD) of RNaseH1 and an engineered variant of ascorbate peroxidase (APEX2) [11]. It allows for the characterization of transient interactions of the proteome with the R-loop in a spatiotemporal manner [141].

As shown in Figure 2A,B, the Immunoprecipitation (IP) and Immunohistochemistry (IHC) of different tissue samples can be used by employing the S9.6 antibody complemented with an R-loop-binding protein antibody in routine diagnostics for preliminary insights.

While several experimental methods are now available to detect, quantify, and study R-loop dynamics, the structural and functional characterization of an R-loop still remains a major challenge for developing new (clinically relevant) therapeutic probes. Along these lines, studies that structurally characterize and target R-loops are currently under development [147] with the help of systematic prediction and detection pipelines. Bioinformatics tools such as QmRLFS-finder, R-loop tracker, and databases such as the R-loop Altas, and R-loop base are next-generation, in silico approaches developed to characterize R-loops. The Quantitative Model of the R-loop-Forming Sequence (RLFS) finder (QmRLFS-finder) [142] is a web-based server that predicts RLFSs based on experimentally supported structural models of RLFSs in RNA/DNA sequences. It demonstrates highly accurate predictions of the RLFSs detected. The R-loop tracker tool is a similar web-based server that focuses on the prediction of R-loops in genomic DNA with an unlimited input size [143]. It allows for the cross-evaluation of in silico results with experimental data, if available, and helps correlate these with other genomic features and markers with an enhanced visualization output [143]. In 2017, R-loopDB, a database that contains computationally predicted RLFSs in human genetic regions, was developed. Using QmRLFS, the updated version of this database now has an increased the number of RLFSs predicted in human and other genomes [145]. It also provides a comprehensive annotation of Ensembl RLFS-positive genes for studying comparative evolution and genome-scale analyses in R-loop biology. The R-loop Atlas is a database harboring about 63 million peaks collected from 254 plant species by ssDRIP-seq and deepR-loopPre (a deep-learning tool for predicting locations and profiles of strand-specific R loops; http://bioinfor.kib.ac.cn/R-loopAtlas/index.html, accessed on 18 November 2022) [144]. R-loop base is another database that includes a reference set of human R-loop zones for high-confidence R-loop localization and for spotting conserved genomic features that are associated with R-loop formation. The data are cured in a comprehensive manner by integrating knowledge from multi-omics analyses and literature mining. A list of R-loop-regulatory proteins and their targeted R-loops in multiple species, to date, can be obtained [148].

Overall, an impressive spectrum of techniques has been developed aiming at elucidating the R-loop distribution genome-wide and its function. These techniques are expected to enrich “clinical” diagnostics and drug discovery in the R-loop field. However, from a diagnostic point of view, there is still a scope for further technical improvements. For example, it would be desirable to acquire techniques that can differentiate between scheduled and unscheduled R-loops, techniques that permit the detection of R-loops from circulating cells, or techniques that allow for R-loop detection with a nucleotide resolution from widely available clinical (paraffin-embedded) material. Furthermore, it would be desirable to develop and advance techniques that allow for the (large-scale) parallel profiling of R-loops in a reliable manner, which in turn may help the identification of disease-specific R-loop-prone loci. Such techniques would foster a more comprehensive understanding of the functional importance and diagnostic potential of R-loops in the context of human disorders. They may also yield complementary biological insights and help us to understand whether R-loop perturbations almost exclusively manifest in pathologies affecting cells with a limited regenerative potential (such as neuronal cells). Given the broad functional spectrum of R-loops in genome organization, gene regulation, and genome integrity, it seems possible that perturbations of R-loop biology could also confer detrimental effects during early (human) development. Such effects are not normally searched for, e.g., in human fetal tissue, and hence the further evolution of technologies used to detect R-loops could provide novel insights into the biology and consequences of scheduled and unscheduled R-loops. Finally, generalized guidelines would be necessary for the clinical use of easy and reproducible techniques.

## 8. Conclusions

Considerable progress has been made in understanding the complexity and regulation of R-loops and their resulting biological functions. Although the perturbations of R-loops have not yet been exhaustively studied beyond the disease entities mentioned in this review, aberrant R-loops appear to trigger or even perpetuate a variety of disorders. R-loops represent a valuable resource that reflects the actual state in real-time and in specific situations, e.g., during development, maturation, and in pathophysiological conditions, making them exciting structures for diagnosis and informing clinical decisions. Burgeoning R-loop-screening technologies are expected to fuel the current and future techniques towards their use in clinical diagnostics and precision medicine.

## Figures and Tables

**Figure 1 genes-13-02181-f001:**
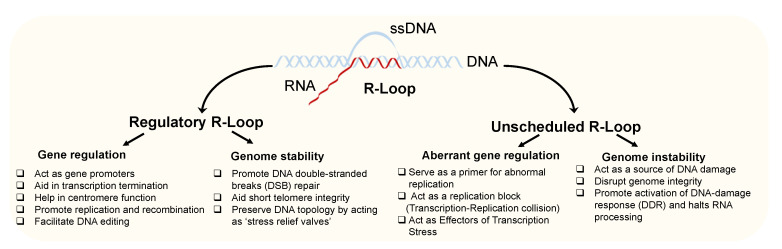
**Roles of Regulatory and Unscheduled R-loops.***Regulatory R-loops are intermediates required for gene regulation and genome stability (**left**).* R-loops regulate gene activity by modulating transcriptional activity, replication, recombination, centromere function, and DNA editing. R-loops are also involved in stabilizing the genome by promoting the repair of DNA double-strand breaks (DSB) and short telomere structures. *Absence of timely removal or prevention of unwanted R-loop accumulation results in the formation of unscheduled R-loops (**right**).* Unscheduled R-loops are a source for aberrant gene regulation (by serving as a primer for aberrant replication, enhancing transcription–replication collision, and acting as an effector of transcription stress) and contribute to the instability of the genome (by inducing DNA damage). In turn, unscheduled R-loop formation promotes the DNA-damage response (DDR) activation and halts RNA processing to allow for genome repair [29].

**Figure 2 genes-13-02181-f002:**
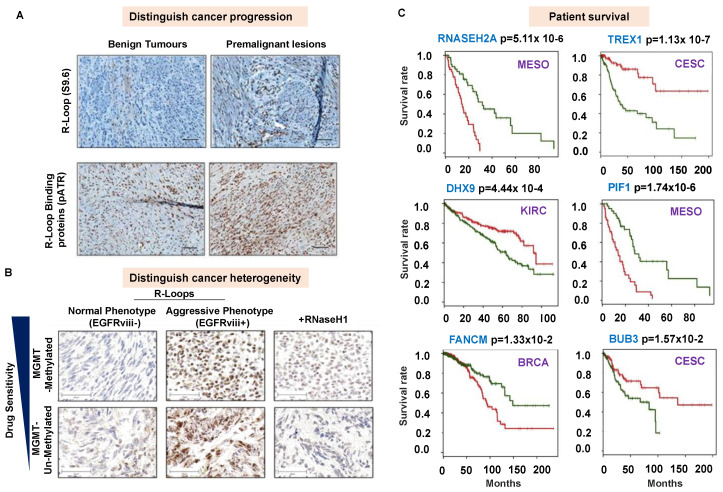
**R-loops and R-loop-binding proteins as biomarkers.** (**A**). *Benign and Premalignant tumor Uterine fibroids are tissue-specifically characterized by aberrant R-loop accrual and its binding protein*. Immunohistochemistry (IHC) staining with (upper panel) R-loop-specific S9.6 and (lower panel) pATR (Ser428)-specific antibodies indicating phosphorylated (activated) replication stress signaling correlative with R loop accumulation in different mutants (Re-adapted from Ref [124]). (**B**). *R loop accumulation used for distinguishing tumor heterogeneity in Glioblastoma (GBM) patients, highlighting distinct biological properties and complementing the readout of established clinical marker for GBM (O6-methylguanine (O6-MeG)-DNA methyltransferase (MGMT))*. IHC staining of nonaggressive versus aggressive phenotype GBM. Epidermal growth factor receptor variant III (EGFRvIII) areas are depicted in the left column, and EGFRvIII+ areas are displayed in the middle column. Aggressive phenotype GBM incubated with RNase H1 (shown in right column) before immunohistochemical staining with S9.6 antibody is used as specificity control (removing R-loops) for staining, indicating that increased EGFRvIII expression is associated with an increase in R loop accumulation predisposed to DNA damage and genomic instability (Re-adapted with Creative Commons CC-BY-NC license permissions from [126]). (**C**). Representative Kaplan–Meier curves showing the overall survival probability in various cancer types (Mesothelioma (MESO), Cervical and endocervical cancers (CESC), Kidney renal clear cell carcinoma (KIRC), and Breast invasive carcinoma (BRCA)) depending on the gene expression of various R-loop-binding proteins (high and low expression shown in red and green, respectively; R-loop genes, cancer IDs, and *p*-values are indicated (Re-adapted from [28]).

**Table 2 genes-13-02181-t002:** **Modulation of R-loop-binding or R-loop modifying proteins by components that control various aspects of gene expression including alternative polyadenylation (APA).** Matrix of selected genes encoding R-loop-binding proteins (*x*-axis) with significant alterations in polyadenylation after depletion of canonical and non-canonical 3′ end-processing factors (*y*-axis—blue boxes indicate significant alterations in polyadenylation after depletion of canonical and non-canonical 3′ end-processing factors; *y*-axis—data obtained from TREND–DB [86] covering a large-scale screening [85] coupled to transcriptome-wide interrogation of alterations in polyadenylation by TREND-seq [89], further details in the text).

		R-Loop Binding Proteins										
			AQR	BUB3	EWSR1	RNASEH1	RNASEH2A	RNASEH2C	THOC5	TREX1	DDX5	EXOSC3
**Transcription**		**YBX1**										
		**CTCF**										
		**LEO1**										
		**XRN2**										
		**SSU72**										
		**POLR2B**										
		**POLR2C**										
**Translation**		**CIRBP**										
**Cleavage and Polyadenylation**	**CPSF**	**CPSF1**										
		**CPSF2**										
		**CPSF3**										
		**CPSF4**										
		**FIP1L1**										
	**CSTF**	**CSTF1**										
		**CSTF2**										
		**CSTF3**										
	**CFIm**	**CPSF6**										
		**CPSF7**										
		**NUDT21**										
	**CFIIm**	**PCF11**										
		**CLP1**										
	**Integrator complex**	**CPSF3L**										
		**SYMPK**										
		**CSTF21**										
		**WDR33**										
		**RBBP6**										
		**CPEB1**										
	**PAP**	**PAPOLA**										
		**PAPOLG**										
	**NC PAP**	**PAPD4**										
		**PAPD5**										
		**PAPD7**										
	**PABP**	**PABPC4**										
		**PABPN1**										
		**PABPC1**										
**RNA processing**		**RBM5**										
		**PTBP1**										
		**DDX39B**										
		**DDX23**										
		**U2AF1**										
		**SCAF1**										
		**SRSF1**										
		**SRSF3**										
		**SRSF4**										
		**RNPS1**										
		**SCAF1**										
		**XRN2**										
		**DIS3L**										
		**SKIV2L2**										
		**DCP2**										
		**ZFP36**										
		**KHSRP**										
		**TARBP2**										
		**HNRNPH1**										
**Epigenetics**		**CHD1**										
		**PARN**										
**Others**		**MAPK9**										

**Table 4 genes-13-02181-t004:** Diagnostic tools used to detect R-loops and R-loop-binding proteins.

	Detection Method	Method Name	Processing Method	Advantages	Disadvantages
Wet lab techniques					
**Techniques for detecting R loop**	**S9.6 antibody staining** **DNA/RNA hybrid**	Immunoprecipitation (IP)/Immunohistochemistry (IHC) [128].	Immunostaining of DNA–RNA hybrid	Good signal, likely useful for the analysis of samples from tissue banks	Limited to the microscopic examination of R- loops
		DRIP [56,70,129,130,131]	Restriction digestion (RE) of genome followed by IP	Robust signal	Better resolution than IP/IHC but is still low
		DRIP-seq [31]	RE of genome followed by IP and dsDNA sequencing	Robust signal is widely adopted, and is easy to set up	Low resolution, no strand specificity, and cannot be used in situ
		S1-DRIP-seq [132]	Sonication of samples followed by IP and dsDNA sequencing	Higher resolution than DRIP-seq	No strand specificity and cannot be used in situ. S1 nuclease is delicate, and it is difficult to control the reaction, which may make it challenging to reproduce the data in clinical setting
		DRIPc-seq [133]	RE of genome followed by RNA sequencing	Strand-specific, high resolution	Not in situ, requires longer sample preparation, S9.6 may recognize dsRNA
		RDIP-seq [134]	Sonication of genome followed by RNA sequencing		Not in situ, tedious preparation
		Bis-DRIP-seq [135]	RE of genome followed by sequencing dsDNA with bisulfite conversions	Strand-specific, provides additional control to ensure S9.6 signal arises from an R-loop in situ	Requires many replicates and shows R-loop enrichment in promoter regions only
		qDRIP [136]	RE of genome followed by IP of DNA–RNA hybrid and synthetic DNA/RNA hybrid used as internal standards followed by dsDNA sequencing and quantification using internal standards as a reference.	Internal standards help with high-resolution, strand-specific sequencing	Spikes in hybrids shorter than 150 bp are unlikely to be useful for normalization. Additional spike-in may be required
		SMRF-seq [137]	RE of genome followed by IP of DNA/RNA hybrid and sequencing of dsDNA with bisulfite conversions at single molecule level	Strand-specific, single-molecule resolution, avoids biases inherent to read-count normalization by accurately profiling signals in regions unaffected by transcription inhibition thus providing accurate differential peak calling between conditions	As with any foot-printing method, SMF is agnostic to the distinguishing of DNA-binding proteincreating the footprints.
	**Catalytically inactive RNase H**	DRIVE-seq [31]	RE of genome followed by targeting catalytically inactive RNaseHs and dsDNA sequencing	Provides independent verification of some DRIP-seq results	Low enrichment, low resolution, reagent not commercially available, no strand specificity, not in situ
		R-ChIP-seq [138]	Sonication followed by targeting catalytically inactive RNaseHs and ssDNA sequencing	Strand specific, in situ capture	Cell line must be engineered to express catalytically inactive RNase H construct, inactive RNase H may alter hybrid dynamics
	**RNase H to guide micrococcal nuclease to R-loops**	MapR [139]	Antibody-independent R-loop-profiling technique that utilizes RNase H to guide micrococcal nuclease to R-loops, which are subsequently cleaved, released, and identified by sequencing	Heavily based on CUT&RUN, a new and fast method to identify transcription factor binding sites genome-wide	Does not discriminate between the template and non-template strands and, therefore, cannot identify which DNA strand is involved in DNA–RNA hybrid formation.
	**Sensor that binds to R loop**	R loop CUT&Tag [140]	Combines CUT&Tag and GST-His6-2×HBD (glutathione S-transferase–hexahistidine–2× hybrid-binding domain) tags as an artificial R loop hybrid sensor to specifically recognize the DNA–RNA hybrids.	Sensitive, reproducible and generates good resolution to sense the R-loop instead of capture strategies that largely contribute to disparities in the previous techniques including R-loop Mapping	Current form of R-loop CUT&Tag does not provide strand information about R loops
**Techniques for detecting R loop and R loop-binding proteins**	Fusion protein that binds to hybrid-binding domain (HBD) of RNaseH1 and an engineered variant of ascorbate peroxidase	RDProx (RNA–DNA Proximity Proteomics) [141]	Provides a snapshot of the R-loop-proximal proteome	In vivo labelling of R-loop-proximal proteins is performed, difficult to solubilize proteins that are amenable to the analysis can be identified, even transient spatiotemporal interactions with low affinity and transient interactions are detected.	Unable to distinguish between direct protein-binding or indirect proteins associated with RNA
	Staining of both DNA–RNA hybrid (S9.6 antibody)+ R-loop-binding proteins (Antibody specific to the desired protein)	IP/IHC (R-loops+ R-Loop-binding proteins) [124]	Immunostaining of DNA–RNA hybrid and their binding proteins	Fast analysis of pathology specimens	Low resolution and limited to routine microscopic analysis
Bioinformatics tools and databases					
**Techniques for detecting R loop**	Structure-based detection and prediction based on existing wet-lab data	QmRLFS-finder [142]	Identifies three structural features of R loop including a short G-cluster-rich region (R-loop initiation zone or RIZ), a structurally non-specified linker (linker), and long downstream region that has high G-density R-loop elongation zone (or REZ) based on experimental data	User-friendly web server and stand-alone tool for rapid and accurate prediction of RLFSs in DNA or RNA sequences shows strong agreement with existing genes and genome-scale experimentally determined R-loops	Information is limited and an updated version needs to be integrated with the growing experimental data
		R loop tracker [143]			
		R-loop atlas [144]	About 63 million peaks called from 254 plant species by ssDRIP-seq and deepR-loopPre are available	User-friendly web server for plants species based on experimental data	Limited to plant species only
**Techniques for R loops and their binding proteins**		R-loop DB [145]	Consists of computationally predicted R-loop-forming sequences (RLFSs) in human genic regions. Using the QmRLFS, the updated version of this database now has an increased number of RLFSs predicted in the human genes and in the genomes of other organisms	Provides comprehensive annotation of Ensembl RLFS-positive genes to study comparative evolution and genome-scale analyses, also R loop-binding proteins	Limited information and an updated version needs to be integrated with the growing experimental data

## Data Availability

Not applicable.

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
