# Peer review of "Pathophysiological Role and Diagnostic Potential of R-Loops in Cancer and Beyond"

_genes, 2022, doi:10.3390/genes13122181_

Round 1
Reviewer 1 Report
This is a well-written comprehensive review highlighting the use of R-loop detection as a diagnostic technique. I recommend this article be published after addressing the following minor comments:
Lines 162-166: The sentence "This is followed by....R-loop mediated TRCs" needs a bit more elaboration as it is unclear how FACT-mediated swapping of nucleosomes around RNA polymerase promotes replication.
Line 176: It would be better to change "critical" to "detrimental".
Line 179: The abrupt jump to replication is difficult to comprehend for a reader new to subject/field. Please simplify the language for the novice.
Paragraph starting line 206: Please edit to indicate which strand of the hybrid do DHX9 and WRN unwind (RNA or DNA).
Line 246: Please consider changing "controls" to "alters" or "disrupts".
Lines 308-309: text repeated. Please delete the repeated words and edit accordingly
I suggest including a line or two in the conclusion to highlight that disease-specific R-loop prone loci have not yet been identified.
It was a pleasure reading this manuscript.
Author Response
We thank both reviewers for acknowledging our work and for helpful suggestions to improve our manuscript. In the following, we provide a detailed point-by-point response to the reviewers’ comments. All changes in the manuscript have been highlighted.
Reviewer 1
This is a well-written comprehensive review highlighting the use of R-loop detection as a diagnostic technique. I recommend this article be published after addressing the following minor comments:
Lines 162-166: The sentence "This is followed by....R-loop mediated TRCs" needs a bit more elaboration as it is unclear how FACT-mediated swapping of nucleosomes around RNA polymerase promotes replication.
Response:
A statement in line 163 highlighting its role in histone deposition and nucleosome assembly facilitating the replication process (PMID: 26804921) has been added.
Line 176: It would be better to change "critical" to "detrimental".
Response:
We thank the reviewer and have amended this aspect in line 174.
Line 179: The abrupt jump to replication is difficult to comprehend for a reader new to subject/field. Please simplify the language for the novice.
Response:
A simplified statement in line 177 has been added.
Paragraph starting line 206: Please edit to indicate which strand of the hybrid do DHX9 and WRN unwind (RNA or DNA).
Response:
Since this is an introductory statement to the paragraph where other helicases are also discussed we did not edit this section at this location. However, a statement in line 224 has been added to indicate that WRN binds to RNA of Okazaki fragment-like hybrids, whereas DHX9 fails to bind to unwind Okazaki fragment-like hybrids suggesting a role in lagging strand maturation of DNA replication PMID: 20385589.
Line 246: Please consider changing "controls" to "alters" or "disrupts".
Response:
We have added “…and sometimes disrupts” (as it refers to several references, where “disrupts” seemed not to be appropriate for all of them) in line 247 now.
Lines 308-309: text repeated. Please delete the repeated words and edit accordingly.
Response:
The repeated text is deleted now in line 312.
I suggest including a line or two in the conclusion to highlight that disease-specific R-loop-prone loci have not yet been identified.
Response:
We thank the reviewer for the suggestion. A statement is now added in line 563.

Reviewer 2 Report
Manuscript Number: genes-1986302
Title: Pathophysiological role and diagnostic potential of R-loops in cancer and beyond
The authors' review proposes the identification of regions of genomic DNA engaged in R-loops as diagnostic potential in multiple diseases. Certainly, it needs to be extended as a diagnostic tool.
Authors with a high numbers of references indicate almost most of the findings in the field.
It is a very clearly written review with such a complex system with a large number of players in different organisms and conditions. Summary tables facilitate access to complex conditions.
However, if the review in intended for a clinical partner, it is completely unfair, since the techniques offered are very delicate (tricky) and produce a large number of artifacts to be used in common clinical conditions.
The authors themselves are contradictory in the descriptions of the data and superficial on several points of the interpretations of the reported work.
Lane 35-36:R-loops are abundant and are estimated to cover up to 5% of the mammalian genome [2].
The estimated 5% is from reference: 2 (a DRIPc-seq was performed on the human embryonic carci-noma Ntera2 cell line) and these are aneuploid cell lines in culture, not normal cell conditions.
Why generalize to all cells?
Moreover, reference 2 is one of the rare articles which sequences the RNA, in the majority of the publications after immunoprecipitation, it is the DNA which is sequenced. It's a question why every lab doesn't sequence RNA? a problem not very well evoked nor discussed in the reviews!!
Lane 80-81: R-loops are structures, which are abundantly found in the promoter regions of genes and are also known as Promoter R-loops.
These are not always cases where authors even cite telomeres, centromeres, etc. These regions are very large in our genome.
These types of sentences introduce an erroneous notion about the structure of R loops and the expected function.
Lane 459-461:It has proved to be a reproducible and consistent method for the identification and localization of R-loops by sequencing the genomic fragments containing DNA-RNA hybrids.
In each publication, only one specific given fragment is represented and it is very difficult to ensure reproducibility from one publication to another, because each time the region analyzed is different. How do authors know they are reproducible? Not discussed, just a phrase without support!!
Lane 466-467: S1-DRIP-seq uses sonication for fragmentation where the displaced ssDNA is removed prior to sonication using S1 nuclease [142].
Any researcher who has practiced S1 nucleases knows that it is very delicate and difficult to control the reaction and difficult to reproduce for clinical use. Citing the work without giving important details is pointless.
499-500:In addition, R-loop CUT&Tag has been established as a sensor-based technique to over-come large discrepancies in R-loop mapping.
Apparently the authors are aware of the discrepancies in the R-loop mapping. Why not discuss them clearly instead of just cataloging the techniques?
Lane 543-544: These techniques are expected to enrich “clinical” diagnostics and drug discovery in the R-loop field.
Several recent references are missing that will propose reproducible technique and easy to use in clinical conditions.
Author Response
Reviewer 2
Title: Pathophysiological role and diagnostic potential of R-loops in cancer and beyond
The authors' review proposes the identification of regions of genomic DNA engaged in R-loops as diagnostic potential in multiple diseases. Certainly, it needs to be extended as a diagnostic tool. Authors with a high numbers of references indicate almost most of the findings in the field.
It is a very clearly written review with such a complex system with a large number of players in different organisms and conditions. Summary tables facilitate access to complex conditions.
However, if the review in intended for a clinical partner, it is completely unfair, since the techniques offered are very delicate (tricky) and produce a large number of artifacts to be used in common clinical conditions.
The authors themselves are contradictory in the descriptions of the data and superficial on several points of the interpretations of the reported work.
Lane 35-36:R-loops are abundant and are estimated to cover up to 5% of the mammalian genome [2].
The estimated 5% is from reference: 2 (a DRIPc-seq was performed on the human embryonic carcinoma Ntera2 cell line) and these are aneuploid cell lines in culture, not normal cell conditions.
Why generalize to all cells?
Moreover, reference 2 is one of the rare articles which sequences the RNA, in the majority of the publications after immunoprecipitation, it is the DNA which is sequenced. It's a question why every lab doesn't sequence RNA? a problem not very well evoked nor discussed in the reviews!!
Response:
We thank the reviewer for highlighting this point. To avoid any discrepancies, we have now removed the R-loop coverage % in line 34.
Lane 80-81: R-loops are structures, which are abundantly found in the promoter regions of genes and are also known as Promoter R-loops.
These are not always cases where authors even cite telomeres, centromeres, etc.
These regions are very large in our genome.
These types of sentences introduce an erroneous notion about the structure of R loops and the expected function.
Response:
We agree with the reviewer and have removed the word “abundant” to avoid further confusion in line 78.
Lane 459-461: It has proved to be a reproducible and consistent method for the identification and localization of R-loops by sequencing the genomic fragments containing DNA-RNA hybrids.
In each publication, only one specific given fragment is represented, and it is very difficult to ensure reproducibility from one publication to another, because each time the region analyzed is different. How do authors know they are reproducible? Not discussed, just a phrase without support!!
Response:
We agree with the reviewer and have removed the sentence to avoid further confusion (line 462).
Lane 466-467: S1-DRIP-seq uses sonication for fragmentation where the displaced ssDNA is removed prior to sonication using S1 nuclease [142].
Any researcher who has practiced S1 nucleases knows that it is very delicate and difficult to control the reaction and difficult to reproduce for clinical use. Citing the work without giving important details is pointless.
Response:
We thank the reviewer for highlighting this important aspect. We have included a statement in line 470 and in the disadvantages column of Table 4.
499-500:In addition, R-loop CUT&Tag has been established as a sensor-based technique to over-come large discrepancies in R-loop mapping.
Apparently, the authors are aware of the discrepancies in the R-loop mapping. Why not discuss them clearly instead of just cataloging the techniques?
Response:
We thank the reviewer for pointing this out. We have listed some of the causes of the discrepancies in the R-loop mapping in the line 506 (PMID: 33597247).
Lane 543-544: These techniques are expected to enrich “clinical” diagnostics and drug discovery in the R-loop field.
Several recent references are missing that will propose reproducible technique and easy to use in clinical conditions.
Response:
We thank the reviewer for pointing this out. The focus of this review is to highlight the (patho) physiological role of R-loops in health and disease, their diagnostic potential as a biomarker, and widely used state-of-the-art techniques for their detection.
The review is intended not only for clinicians but also for basic researchers. It aims to lay the foundation for future developments to address unmet needs and exploit the untapped diagnostic potential of R-loops in health and disease. We have comprehensively discussed the techniques widely used in the field. As the reviewer highlighted in his/her previous comments, there are discrepancies in the reproducibility of each technique, and we do not want to give an erroneous notion of cataloguing the best possible techniques for clinical use. We feel that this is beyond the scope of this review. However, to acknowledge this important aspect, we have added a statement in line 564 that a general guideline for the use of R-loop techniques for clinical use.

Round 2
Reviewer 2 Report
I thank you author to introduce in the review almost all my suggestions.
I suggest to the editor to accept now their manuscript.